# Malondialdehyde-Acetaldehyde Modified (MAA) Proteins Differentially Effect the Inflammatory Response in Macrophage, Endothelial Cells and Animal Models of Cardiovascular Disease

**DOI:** 10.3390/ijms222312948

**Published:** 2021-11-30

**Authors:** Michael J. Duryee, Dahn L. Clemens, Patrick J. Opperman, Geoffrey M. Thiele, Logan M. Duryee, Robert P. Garvin, Daniel R. Anderson

**Affiliations:** 1Department of Internal Medicine, University of Nebraska Medical Center, 982650 Nebraska Medical Center, Omaha, NE 68198-2650, USA; mduryee@unmc.edu (M.J.D.); dclemens@unmc.edu (D.L.C.); patrick.opperman@unmc.edu (P.J.O.); gthiele@unmc.edu (G.M.T.); logan.duryee@unmc.edu (L.M.D.); robbie.garvin@unmc.edu (R.P.G.); 2Veterans Affairs (VA) Nebraska-Western Iowa Health Care System, 4101 Woolworth Avenue, Omaha, NE 68105-1850, USA

**Keywords:** atherosclerosis, inflammation, oxidative stress, malondialdehyde, acetaldehyde, protein modification, protein adduction

## Abstract

Chronic inflammation plays a critical role in the pathogenesis of atherosclerosis. Currently, the mechanism(s) by which inflammation contributes to this disease are not entirely understood. Inflammation is known to induce oxidative stress, which can lead to lipid peroxidation. Lipid peroxidation can result in the production of reactive by-products that can oxidatively modify macromolecules including DNA, proteins, and lipoproteins. A major reactive by-product of lipid peroxidation is malondialdehyde (MDA). MDA can subsequently break down to form acetaldehyde (AA). These two aldehydes can covalently interact with the epsilon (ε)-amino group of lysines within proteins and lipoproteins leading to the formation of extremely stable, highly immunogenic malondialdehyde/acetaldehyde adducts (MAA-adducts). The aim of this study was to investigate the inflammatory response to MAA-modified human serum albumin (HSA-MAA) and low-density lipoprotein (LDL-MAA). We found that animals injected with LDL-MAA generate antibodies specific to MAA-adducts. The level of anti-MAA antibodies were further increased in an animal model of atherosclerosis fed a Western diet. An animal model that combined both high fat diet and immunization of MAA-modified protein resulted in a dramatic increase in antibodies to MAA-adducts and vascular fat accumulation compared with controls. In vitro exposure of endothelial cells and macrophages to MAA-modified proteins resulted in increased fat accumulation as well as increased expression of adhesion molecules and pro-inflammatory cytokines. The expression of cytokines varied between the different cell lines and was unique to the individual modified proteins. The results of these studies demonstrate that different MAA-modified proteins elicit unique responses in different cell types. Additionally, the presence of MAA-modified proteins appears to modulate cellular metabolism leading to increased accumulation of triglycerides and further progression of the inflammatory response.

## 1. Introduction

Cardiovascular disease (CVD) is the leading cause of death worldwide [1]. The majority of these deaths are caused by coronary artery disease (CAD), which is a consequence of advanced atherosclerosis [1]. The pathogenesis and progression of atherosclerosis is not fully understood, but chronic inflammation is recognized as being instrumental in its initiation and progression [2,3,4]. Although chronic inflammation has a critical role in atherosclerosis, the mechanism(s) that drives chronic inflammation is not completely understood. 

Many of the risk factors associated with CVD including hypertension, hyperlipidemia, smoking, diabetes, and obesity are associated with increased levels of reactive oxygen species (ROS) [5,6]. ROS can damage cells by binding to macromolecules such as DNA, lipids, and proteins. If the level of ROS exceeds the capacity of the cell to neutralize them, a state of oxidative stress develops. Oxidative stress can cause inflammation; likewise, inflammation can cause oxidative stress. Thus, these processes are intimately related and can establish a self-perpetuating cycle contributing to chronic inflammation [7,8].

Oxidative stress can result in lipid peroxidation. One of the major by-products of lipid peroxidation is malondialdehyde (MDA) [9,10,11]. MDA is a mediator and marker of inflammation and has been associated with atherosclerosis [12,13]. MDA can break down to form another reactive aldehyde, acetaldehyde (AA) [14]. MDA and AA can covalently interact with the epsilon (ε)-amino group of lysines within proteins and lipoproteins forming highly stable and immunogenic malondialdehyde-acetaldehyde adducts (MAA-adducts) [15]. MAA-adducts are extremely immunogenic and both polyclonal and monoclonal antibodies have been developed that are highly specific for epitopes of MAA-modified proteins [15,16,17]. Furthermore, in the absence of adjuvant they cause a robust specific adaptive immune response to the MAA structure, the MAA-modified macromolecule, and the carrier structure of MAA-adducted macromolecules [18,19]. Of clinical importance, we have shown that MAA-adducts are present in aortic atheroma from experimental animals and human CVD patients [15,20]. Furthermore, we have shown that antibody titers to MAA-adducts are associated with disease severity and predictive of CVD [18,19,20,21]. Because of these findings, we propose that MAA-modified proteins and lipoproteins play an important pathogenic role in the development of atherosclerosis.

Based on studies in animals and observation of human samples it is generally thought the initial event in the development of an atherosclerotic lesion is damage to the vascular endothelium. Damage or irritation to the endothelial cells by classic CVD risk factors or pro-inflammatory molecules results in activation of the endothelial cells and expression of adhesion molecules. Leukocyte adhesion to the endothelial wall is facilitated by expression of these adhesion molecules. Concomitant changes in endothelial permeability promotes migration of monocytes and retention of cholesterol-containing low-density lipoprotein (LDL) and oxidatively modified cholesterol containing LDL (oxLDL) particles in the artery wall. Specific receptors on the surface of the monocytes result in the endocytosis of both LDL and oxLDL leading to the intracellular accumulation of cholesterol in these cells [22]. Finally, chemoattractants direct the migration of the monocytes to the intima, the innermost layer of the artery wall, where these monocytes differentiate into macrophages. Continued endocytosis of cholesterol laden LDL and oxLDL particles by macrophages result in the formation of foam cells. Accumulation of cholesterol containing foam cells leads to the formation of a fatty streak and the initiation an atherosclerotic lesion. However, it is not known whether MAA modified proteins on LDL will generate foam cells similar to oxLDL. 

The aim of this study was to investigate the effects of MAA-modified proteins in processes important in the development of atherosclerosis. To further our understanding of the role that MAA-adducts play in atherosclerosis, we investigated the effects of MAA-modified proteins in cultured endothelial cells and macrophages. Additionally, we investigated the effects of immunizing experimental animals with MAA-modified proteins. By better understanding the inflammatory response to MAA-modified proteins, we hope to increase our understanding of the role of inflammation in the pathogenesis of atherosclerosis.

## 2. Results

### 2.1. Proteins on LDL Are MAA Adducted

Studies to determine to what extent LDL could be MAA-modified were initiated. MAA-modification of LDL was very similar to that of HSA, achieving a fluorescence of approximately 20,000 arbitrary units at 72 h (Figure 1A). SDS page and immunoblot analysis were performed to determine the extent of modification. Silver stain of proteins demonstrated no drastic shift in the banding pattern of LDL when modified with MAA, indicating no apparent major changes to the proteins (Figure 1B). Both mouse monoclonal and rabbit polyclonal antibodies directed against MAA-modified proteins reacted with LDL-MAA confirming that LDL was MAA-modified with the predominate modified protein appearing to be APO-B100 (Figure 1B).

### 2.2. MAA Modified LDL Is Immunogenic

Studies were then initiated to determine if LDL-MAA was immunogenic in a mouse model. BALB/c mice were immunized with LDL or LDL-MAA as described for animal study 1, and sera analyzed by ELISA for the presence of anti-MAA antibodies using HSA and HSA-MAA as the targets. A significant (*p* < 0.0001) increase in anti-MAA antibodies was detected in the mice immunized with LDL-MAA (Figure 2A). 

To determine if anti-MAA antibodies were present without inoculation of modified proteins, (similar to what has been observed in humans with CVD) an animal model of CVD using Sprague Dawley and JCR rats fed a Western diet was implemented. After 6 months of feeding, serum was collected and analyzed for anti-MAA antibodies using both HSA-MAA and LDL-MAA as targets. Sprague Dawley rats developed significant (*p* < 0.01) levels of antibodies to both LDL-MAA and HSA-MAA (Figure 2B). The JCR rats developed significantly higher (*p* ≤ 0.001) antibody levels to both LDL-MAA and HSA-MAA compared with the unmodified proteins. Importantly, the antibody levels to HSA-MAA (*p* ≤ 0.001) and LDL-MAA (*p* ≤ 0.01); detected in the JCR rats were significantly higher than those detected in the Sprague Dawley rats (Figure 2B). These in vivo studies demonstrate that both injection of MAA-modified protein and feeding of Western diet to animals increases the antibody levels to MAA-adducted proteins.

### 2.3. LDL-MAA Increases Fat Accumulation in Vascular Endothelial Cells

To determine the biochemical effects of MAA-modified proteins, we cultured mouse vascular endothelial cells (CRL-2167) in the presence of LDL-MAA. CRL-2167 cells incubated with LDL-MAA demonstrated increased accumulation of fat as assessed by Oil Red O staining (Figure 3A,C). Quantification of the staining demonstrated that treatment with LDL-MAA resulted in a significant increase in fat accumulation compared with control cells (*p* ≤ 0.001) or cells treated with unmodified LDL (*p* ≤ 0.001) (Figure 3B). To confirm these findings, cellular levels of triglycerides were determined for the three treatments. A significant (*p* ≤ 0.001) increase in triglycerides was detected in endothelial cells treated with LDL-MAA compared with control cells or cells treated with LDL (Figure 3D).

### 2.4. MAA Adducted Proteins Increase Adhesion Molecules and Cytokines in Vascular Endothelial Cells 

We next determined if MAA-modified proteins increased the expression of adhesion molecules and cytokines involved in atherosclerosis using CRL-2167 cells. Analysis of the data showed a significant increase in ICAM-1 (*p* ≤ 0.001) and VCAM-1 (*p* ≤ 0.001) in cells cultured in the presence of HSA-MAA compared with HSA, LDL, or LDL-MAA (Figure 4A,B). Analysis of the expression of monocyte chemoattractant protein-1 (MCP-1) showed treatment with HSA-MAA resulted in a significant (*p* ≤ 0.01) increase. Additionally, treatment with LDL-MAA demonstrated a highly significant increase in the expression of this cytokine (*p* ≤ 0.001) (Figure 4C). Expression of both interleukin-6 (IL-6) and tumor necrosis factor-alpha (TNF-α) were significantly increased in response to HSA-MAA (*p* ≤ 0.001) (Figure 4D,E).

We next investigated the cell surface expression of proteins in response to MAA modification. ICAM-1 expression was significantly (*p* ≤ 0.001) increase in cells treated with either HSA-MAA or LDL-MAA compared with the unmodified proteins (Figure 4F). Similarly, VCAM-1 expression was significantly increased in cells treated with HSA-MAA (*p* < 0.0001) and LDL-MAA (*p* < 0.0001) compared with the respective unmodified proteins (Figure 4G). Additionally, the surface expression of VCAM-1 was significantly increased in LDL-MAA treated cells compared with cells treated with HSA-MAA (*p* < 0.0001) (Figure 4G). Furthermore, treatment of cells with LDL-MAA resulted in a significant increase in the surface expression of the T-cell co-stimulation molecule, CD86 (*p* < 0.0001) compared with all other treatments (Figure 4H).

### 2.5. LDL-MAA Increases Fat Accumulation in Mouse Macrophage

To investigate the effects of MAA-modified proteins in macrophages, we used the murine monocyte/macrophage cell line, J774. Qualification of the Oil Red O staining revealed significantly more fat in cells treated with LDL-MAA compared with either untreated or LDL treated cells (*p* < 0.001) (Figure 5A,B). To confirm the staining data, triglyceride levels were determined for the three treatments. A significant (*p* < 0.001) increase in triglycerides was detected in J774 cells treated with LDL-MAA compared to no treatment or treatment with LDL (Figure 5C). 

### 2.6. MAA-Modified Proteins Increase Pro-Inflammatory Cytokines in Macrophage

Cytokines play a major role in the communication between macrophages and other cells involved in the immune response. Therefore, using RT-PCR, J774 cells were analyzed for their cytokine response to MAA-modified proteins. Analysis of IL-6 expression revealed that treatment with HSA-MAA resulted in a significant increase in expression (*p* < 0.001) compared with unmodified HSA. No increase was observed in cells treated with LDL-MAA (Figure 6A). Similarly, interleulin-1β (IL-1β) expression was significantly increased in HSA-MAA treated cells (*p* < 0.0001); whereas, no significant changes were observed in cells cultured in the presence of LDL-MAA (Figure 6C). Analysis of TNF-α and MCP-1 levels revealed that treatment with HSA-MAA significantly increased their expression compare with unmodified HSA (*p* < 0.0001) (Figure 6C,D). Additionally, treatment with LDL-MAA lead to a significant increase in TNF-α expression (*p* < 0.01). Surprisingly, treatment with unmodified LDL resulted in a slight, although significant increase (*p* < 0.01) in expression of MCP-1 compared with LDL-MAA (Figure 6C). These data indicate that the carrier protein influenced the cytokine expression response.

### 2.7. High Fat Diet and MAA Antigen Increase Aortic Fat Deposition (Key Finding)

We next determined if injection of MAA-modified protein in combination with a high fat diet would increase the immunological response to MAA-modified proteins. Rats were fed either a control or a high fat diet and immunized with HSA-MAA. Antibodies to MAA-modified protein were significantly increased in the sera of rats fed a high fat diet and immunized with HSA-MAA (*p* < 0.0001) compared with HSA-MAA immunized rats fed a normal chow diet (Figure 7A). Although the absolute levels were comparatively much lower than in the immunized animals, rats fed a high fat diet but not immunized with HSA-MAA had higher levels of antibodies to MAA-modified protein than non-immunized rats fed normal chow (Figure 7B), which was observed in Figure 2B.

To determine if the increase in antibody levels to MAA-modified proteins was associated with pathologic changes, we investigated fat accumulation in the aortas of these rats. The results demonstrated that aortas from rats fed control diet (regardless of immunization of HSA-MAA) had virtually no lipid accumulation (Figure 7C). The aortas of non-immunized animals maintained on a high fat diet showed slight accumulation of fat along the inner wall, of the aorta. In contrast, rats immunized with HSA-MAA and maintained on a high fat diet contained numerous lipid deposits both on the inner wall and in the vasavasorum of the aorta. Quantification of the Oil red O staining showed a significant increase in Oil red O staining in animals fed the high fat diet (*p* < 0.01) compared with control-fed or MAA-injected control-fed rats. Furthermore, the aortas from the MAA-immunized rats maintained on a high fat diet contained the highest levels of fat (*p* < 0.0001) (Figure 7D). This same aortic tissue was stained for MAA antigen and revealed strong reactivity to MAA in the animals fed high fat and immunized with the HSA-MAA (Figure 7E) with a significant (*p* < 0.0001) increase for integrated density (Figure 7F).

## 3. Discussion

Atherosclerosis is a disease driven by chronic low-grade inflammation and aberrant lipid metabolism [23]. The microenvironment in which atherosclerosis occurs is characterized by cellular responses, including cytokine and chemokine secretion, uptake of modified proteins and lipoproteins, as well as cellular transformation that ultimately result in the formation of atherosclerotic plaques. Inflammation can cause oxidative stress, which in turn can lead to further inflammation; thereby, establishing a self-perpetuating cycle that can lead to a state of chronic inflammation [7]. It is thought that the oxidation of lipids, proteins and lipoproteins contained in LDL have an important role in the development of atherosclerosis [24]. One of the oxidative modifications that occurs to these macromolecules is irreversible MAA-modification. MAA-modified proteins have been detected in atherosclerotic lesions [20,21,25]. Because MAA-adducts are by-products of oxidation and highly immunogenic, their presence in these lesions indicate that, they may play an important role in the development of this disease. MAA-adducts are highly immunogenic and elicit an immune response, we propose that they may have a role in the perpetuation of the cycle of inflammation and oxidative stress and in this way contribute to the maintenance of the chronic inflammatory state of atherosclerosis.

Two cell types involved in the early stages of atherosclerosis are endothelial cells and monocytes. The initial event associated with atherosclerosis is thought to be damage to the vascular endothelium. The damaged endothelial cells secrete cytokines and chemokines and increase their expression of the adhesion molecules, ICAM-1 and VCAM-1 [3]. Treatment of the vascular endothelial cell line CRL-2167 with HSA-MAA increased the mRNA expression of both ICAM-1 and VCAM-1. Analysis of the surface expression of ICAM-1 and VCAM-1 revealed that both HSA-MAA and LDL-MAA significantly increased the cell surface expression of these adhesion molecules. The increased surface expression of these adhesion molecules may contribute to atherogenesis as it has been shown that expression of ICAM-1 is critical in the early stages of atherogenesis, and expression of VCAM-1 is important in the later stages of the atherogenic process [26]. Furthermore, our data demonstrate that treatment of the vascular endothelial cells CRL-2167 with HSA-MAA significantly increased the expression of the pro-inflammatory cytokines, MCP-1, IL-6, and TNF-α. Interestingly, treatment of CRL-2167 cells with LDL-MAA only increased MCP-1 expression. This differential response could be attributed to binding to multiple different scavenger receptors, driving very different unique responses. 

Atherosclerosis progresses when monocytes attached to the damaged vascular endothelium migrate into the intima, the inner most layer of the artery [27]. Once present in the intima, monocytes accumulate lipids and differentiate into macrophages. If accumulation of lipids continues, the morphology of the lipid containing macrophages changes, eventually taking on a foamy appearance and become what are termed foam cells. LDL particularly, cholesterol laden LDL is the major source of lipid that accumulates in these cells [28]. Interestingly, we found treatment of both the endothelial cell line CRL 2167 and the monocyte/macrophage cell line J774, with LDL-MAA resulted in the accumulation of triglycerides.

Further investigation of the effects from MAA-modified proteins on the J774 cells revealed that treatment of these cells with HSA-MAA significantly increased the expression of the pro-inflammatory cytokines IL-6, IL-1β, TNF-α, and MCP-1. Treatment with LDL-MAA increased the expression of TNF-α compared with cells treated with unmodified LDL. These data indicate that MAA-modified proteins elicit a pro-inflammatory response in monocytes/macrophages. Importantly, there was a differential cytokine response in cells treated with HSA-MAA or LDL-MAA. Thus, it appears that the carrier protein influences the cytokine response and may drive the activation of very different pathways. However, it would be possible in the circulation for both albumin and LDL to become MAA modified and generate their individual unique responses or exacerbate the response together. These in vitro data highlight three important points: (1) treatment of endothelial cells and monocyte/macrophages with MAA-modified proteins alters their metabolism in such a way as to result in the intracellular accumulation of lipids, (2) treatment of these cells with MAA-modified proteins induces the expression of pro-inflammatory cytokines, and (3) specific MAA-modified proteins elicit different cytokine responses. 

Oxidized proteins, including MAA-modified proteins, have been shown to be recognized on the cell surface by scavenger receptors [29,30,31]. Scavenger receptors are a heterogeneous group of receptors that bind, internalize, and transport modified proteins [32]. Different scavenger receptors bind different ligands. For example, one subset of scavenger receptors binds to oxidized LDL; whereas, another subset of scavenger receptors binds acetylated LDL [32]. Binding to scavenger receptors initiates a signal transduction response that likely initiates the various scavenger receptors signaling cascades [33]. Additionally, many MAA-modified proteins likely bind to more than one scavenger receptor and the various combinations of bound receptors direct different signaling responses [18,30,31]. The binding of modified proteins to specific scavenger receptors may be influenced by the specific protein modified, as well as by the specific modification. Our data indicates that it is likely that the specific protein that is modified affects the subset of receptors that is bound. Subsequently, this directs the signaling events that lead to the up-regulation of cytokines and a specific inflammatory response. The differential results seen between HSA-MAA and LDL-MAA stimulation of cells could be attributed to that fact that one is a soluble protein and the other is a lipid protein. Lipids likely get into the cell without signaling, while proteins bind and signal in a proinflammtory nature. More work will need to be done to determine the role of these modified proteins and lipids. 

Not only are MAA-modified proteins generated in vivo they also have important in vivo effects. (Key Finding) Rats feed a Western diet containing high cholesterol developed higher anti-MAA titers. This is presumably because increased dietary cholesterol provides increased levels of substrate for lipid peroxidation, which leads to subsequent increased MAA-adduct production. Additionally, immunization of rats maintained on a high fat diet with HSA-MAA further enhanced the anti-MAA response indicating that MAA-modified proteins are immunogenic, and an oxidative environment provided by the high fat diet, can enhance the production of anti-MAA antibodies and in this way increase the magnitude of the immune response. Although repeated immunization with HSA-MAA in the absence of a high fat diet resulted in increased antibody titers to MAA-adducts, immunization with HSA-MAA alone did not result in readily observable changes in the cardiovasculature. Conversely, feeding the rats a high fat diet resulted in a modest accumulation of fat in the aorta. When immunization with HSA-MAA was combined with administration of a high fat diet, a dramatic accumulation of fat was observed in the aorta. MAA antigen was present near this fat accumulation, indicating there may be a bigger involvement for the role of this adduct. Thus, it appears that a high fat diet causes minimal damage and that the increased levels of MAA-modified proteins or perhaps the increased immune response to MAA-adducts in concert with a high fat diet leads to more severe pathological changes to the vasculature. This unique finding would suggest that MAA modified proteins increase fat uptake around the vessel, leading to potential plaque build-up, eventually leading towards formation of plaque. Future studies to determine if we can prevent this fat formation with drugs that scavenge MAA prior to its formation are currently underway. There is strong evidence that both methotrexate and doxycycline could prevent this formation of MAA induced fat accumulation [34].

In conclusion, MAA-modified proteins induce the formation of lipid laden endothelial cells and macrophages. Additionally, MAA-modified proteins elicit a response characterized by increased expression of cytokines, and the upregulation of adhesion molecules. These actions may help perpetuate the inflammatory response and fuel the initiation and progression of atherosclerosis. MAA-modification of both LDL and HSA can drive antibody production, which is dramatically increased with the addition of a high cholesterol or high fat diet. Proteins and lipoproteins in the atherosclerotic lesion can then become MAA-modified and drive the self-perpetuating cycle of events that eventually lead to plaque formation and occlusion of the vessels in the heart. These data provide new insight into the role of MAA-modification of lipoproteins and proteins in cardiovascular disease and may help to develop new strategies to prevent or treat atherosclerosis. This inflammatory response may lead to the recruitment of monocytes and their migration to the intima. Additionally, the fact that LDL-MAA induces accumulation of fat in these cells indicates that MAA-modification may also have a role in the accumulation of fat in monocytes and the development of foam cells.

## 4. Materials and Methods

### 4.1. Malondialdehyde-Acetaldehyde (MAA)-Protein Adduct Formation

Malondialdehyde-acetaldehyde-modified proteins were produced by reacting 1 mg/mL of HSA (pharmaceutical grade HSA, (Talecris Biotherapeutics Inc., Research Triangle Park, NC, USA) or 1 mg/mL of LDL (Alfa Aesar, Ward Hill, MA, USA) with 2 mM MDA, and 1 mM AA, in phosphate-buffered saline, for 72 h at 37 °C. Formation of MAA-modified proteins was monitored by their auto-fluorescence (excitation 398 nm and emission 460 nm) in a Turner Biosystems (Sunnyvale, CA, USA) LS-5B spectrofluorometer as previously described [15]. SDS page gel electrophoresis and immunoblot analysis was performed on the modified proteins to determine changes in protein confirmation and potential proteins within LDL that were MAA-modified.

### 4.2. Animal Studies

Three separate animal studies were performed. Animal study 1; naïve female BALB/c mice were purchased from Jackson Laboratory (Bar Harbor, ME, USA), fed a normal mouse chow diet, and inoculated intraperitoneal (IP) with 25 µg of human LDL or LDL-MAA weekly for 5 weeks. Mice were euthanized and sera and tissues collected. Animal study 2; male JCR (leptin receptor knockout) obese diabetic atherosclerotic rats and Sprague Dawley rats were purchased from Charles Rivers Laboratories (Wilmington, MA, USA). The JCR rats and Sprague Dawley rats were fed a high cholesterol, “Western” diet for 6 months. After the six-month feeding period, the animals were euthanized and sera and tissues collected for analysis. Animal study 3; male Sprague Dawley rats were purchased from Charles Rivers Laboratories and divided into four groups: (1) control diet, (2) high fat diet (1.25% cholesterol, 0.5% cholic acid, 17% cocoa butter, 0.5% 2-thiouracil) (Research Diets Inc, New Brunswick, NJ, USA), (3) control diet + injection with 100 μg/mL HSA-MAA, and (4) high fat diet + injection with 100 μg/mL HSA-MAA. Rats were fed for a total of 12 weeks, at week 6 rats were injected with HSA-MAA every week for the remaining of the feeding time period. At week 12, the rats were sacrificed, and sera and aortic tissue collected and analyzed. Serum anti-MAA antibody levels were determined by enzyme-linked immunosorbent assay (ELISA) as previously described in all three animal studies [16]. All procedures were approved by the IACUC at the University of Nebraska Medical Center in accordance with the National Institutes of Health Guide for the Care and Use of Laboratory animals. 

### 4.3. Cell Culture

Murine monocyte/macrophage cells (J774) and vascular endothelial cells (CRL 2167) [25,26] were cultured in high glucose Dulbecco’s Modified Eagles Medium supplemented with, 44 mM sodium bicarbonate, 2 mM L-glutamine, 50 micrograms/mL gentamicin, and 10% fetal bovine serum (FBS) (10% complete media). 

### 4.4. Detection of Cellular Lipids

J774 and CRL 2167 cells were cultured in four well slide chambers for 24 h. Media was changed to serum free media containing 50 μg/mL LDL or LDL-MAA and the cells were incubated for 24 h. Cells were then washed with PBS, chambers removed, and the slides fixed with 10% formalin for 1 h. Isopropanol was added to a final concentration of 60% and slides were incubated for 5 min. The isopropanol was removed, and the slides were stained with Oil Red O (working solution 0.3%) for 15 min. Slides were washed in distilled water 5 times, and overlaid with cover slips. Light microscopy was performed to determine lipid accumulation. This technique was also used to determine fat accumulation in the aortas of rats. Triglycerides were quantified in J774 and CRL 2167 cells after incubation with 50 μg/mL LDL or LDL-MAA for 48 h using a triglyceride assay kit (Cayman Chemical, Ann Arbor, MI, USA) as per manufacturer recommendations. 

### 4.5. RT-PCR 

Cells were seeded on to 24 well plates and treated with 25 µg/mL for 6 h. Following treatment, culture media was removed, and the cells were processed for RNA isolation. Briefly, the cells were washed, lysed, and RNeasy (Qiagen, Germantown, MD, USA) kits were used to isolate the RNA according to the manufacturer’s instructions. RNA concentrations were determined and reverse transcribed using the High-Capacity cDNA Reverse Transcription Kit (Applied Biosystems, Carlsbad, CA, USA). RT-PCR was performed using an Applied Biosystems 7500 Real-Time PCR system to determine mRNA expression of pro-inflammatory cytokines. The levels of expression were assessed by the 2-∆∆CT method using GAPDH as an internal reference.

### 4.6. Flow Cytometry

CRL-2167 cells were cultured as described above and treated with 25 µg/mL of HSA, HSA-MAA, LDL, or LDL-MAA for 24 h. Cells were then washed and stained with the following antibodies: mouse anti-mouse BUV395 labeled intercellular adhesion molecule-1 (ICAM-1) (BD Biosciences, San Jose, CA, USA), rat anti-mouse VioBright 515 labeled vascular cell adhesion molecule (VCAM-1) (Miltenyi Biotec, Auburn, CA, USA), or rat anti-mouse Brilliant Violet 510 labeled anti-CD86 (Biolegend, San Diego, CA, USA). Compensation beads were used to correct for spectral overlap and cells were stained for vitality using a LIVE/DEAD cell stain (Invitrogen, Carlsbad, CA, USA). Dead cells were gated out of the analysis and data is expressed as percent positive compared with the antibody controls.

### 4.7. Immunohistochemistry

Paraffin embedded aortic tissue was subjected to antigen retrieval, block with goat serum and incubated with a Zenon Alex Fluor 694 label (Invitrogen) polyclonal rabbit anti-MAA antibody. DAPI (4′,6-diamidino-2-phenylindole; to identify nuclei) was added and samples were sealed with Fluormount-G (Southern Biotech, Birmingham, AL, USA). Fluorochrome detection was done using a Revolve fluorescent microscope (ECHO, San Diego, CA, USA). Images were quantified using ImageJ (NIH, Bethesda, MD, USA).

### 4.8. Statistical Analysis

All data are expressed as the mean ± SEM. Statistical analysis was performed using GraphPad Prism (GraphPad Software, San Diego, CA, USA). Significance was assessed using one-way or multiple ANOVA with Bonferroni’s multiple comparisons test where appropriate. Differences were considered statistically significant at *p* ≤ 0.05. Analysis of oil red O images was done using ImageJ software from the U.S. National Institutes of Health [35].

## Figures and Tables

**Figure 1 ijms-22-12948-f001:**
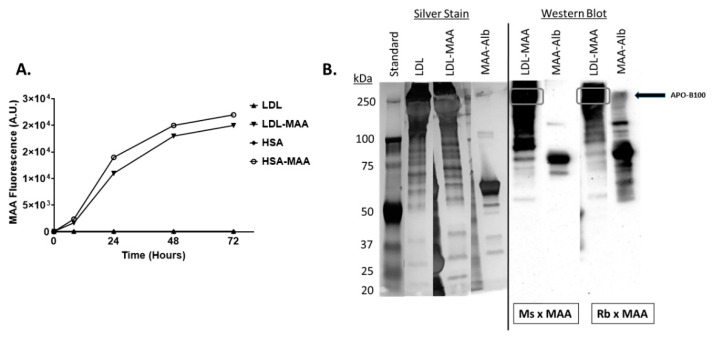
MAA-Modification of LDL and HSA. (**A**). The formation of MAA-modified proteins was monitored by its fluorescence (excitation 398 nm and emission 460 nm). Data is expressed in arbitrary units (A.U.) of MAA Fluorescence in scientific notation. (**B**). Modified proteins were analyzed by polyacrylamide gel electrophoresis. The gels were silver stained or immunoblotted and probed with anti-MAA-adduct mouse monoclonal or rabbit polyclonal purified antibodies to identify potential modified proteins. Silver stained gels were exposed for 30 s to detect banding patterns, while Western Blots were exposed for 1–2 min using chemiluminescence reagent.

**Figure 2 ijms-22-12948-f002:**
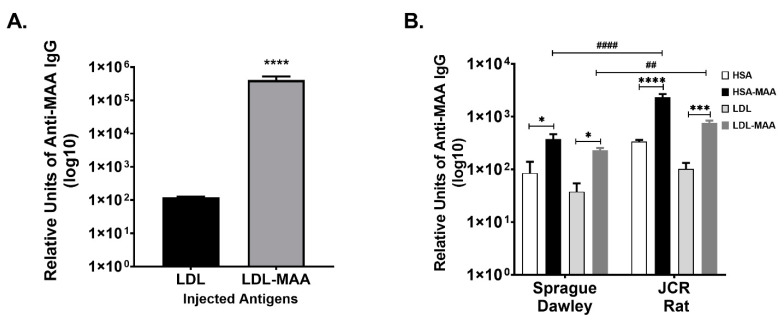
In Vivo Production of Antibodies to MAA-Modified Proteins. (**A**). Female BALB/C mice injected with LDL or LDL-MAA and sera screened against HSA-MAA and HSA as the targets. Data is expressed as the scientific notation log^10^ in relative units of anti-MAA IgG antibody. **** *p* ≤ 0.0001 compared to LDL injected mice. *N* = 5 per group. (**B**). Male Sprague Dawley and JCR rats were fed a high cholesterol diet for 6 months. Sera was collected and analyzed by ELISA for the presence of antibodies to HSA, HSA-MAA, LDL or LDL-MAA. Data is expressed as the scientific notation log^10^ in relative units of anti-MAA IgG antibody. Significance: * *p* ≤ 0.01 serum antibodies to HSA-MAA and LDL-MAA were increased compared to HSA or LDL controls in Sprague Dawley rats. **** *p* ≤ 0.0001 serum antibodies to HSA-MAA were increased compared to HSA in JCR rats. *** *p* ≤ 0.0001 serum antibodies to LDL-MAA were increased compared to LDL in JCR rats. * *p* ≤ 0.01 serum antibodies to HSA-MAA and LDL-MAA were increased compared to HSA or LDL controls in Sprague Dawley rats, #### *p* ≤ 0.01 antibodies to HSA-MAA were significantly increased in JCR rats compared to Sprague Dawley ## *p* ≤ 0.01 antibodies to LDL-MAA were significantly increased in JCR rats compared to Sprague Dawley. N = 5 animals per group.

**Figure 3 ijms-22-12948-f003:**
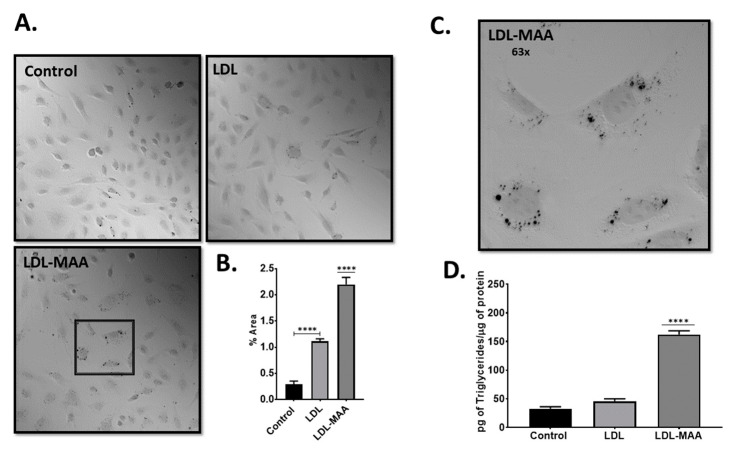
Fat Accumulation in CRL 2167 Vascular Endothelial Cells after LDL-MAA Treatment. (**A**). Representative 10x microscopic images from cells stained with Oil Red O. The box within the LDL-MAA image depicts the region taken for figure C. (**B**). ImageJ quantification of images. Data is expressed as the mean pixel density of the percent area determined for the entire slide. (**C**). Representative higher magnification 63x image of CRL 2167 cells containing lipid droplets. (**D**). Quantification of triglycerides. Data is expressed as the pictograms of triglycerides per microgram of cellular protein. Significance: **** *p* ≤ 0.0001. N = 6.

**Figure 4 ijms-22-12948-f004:**
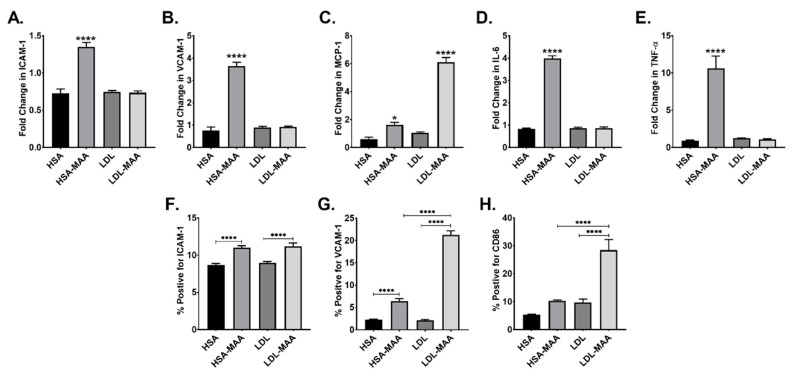
Effects of Treatment of CRL 2167 Vascular Endothelial Cells with MAA-Adducted Proteins. RT-PCR for the expression of pro-inflammatory cytokines and adhesion molecules. (**A**). ICAM-1, (**B**). VCAM-1, (**C**). MCP-1, (**D**). IL-6, (**E**). TNF-α. Data is expressed as the fold change in mRNA compared to treatment with media only. (**F**). Flow cytometric analysis of cells stained for ICAM-1, (**G**). Flow cytometric analysis of cells stained for VCAM-1, (**H**). Flow cytometric analysis of cells stained for CD86. Data is expressed as the percent positive cells for the given cell surface marker compared to treatement with media only. Significance: **** *p* ≤ 0.0001, N = 8.

**Figure 5 ijms-22-12948-f005:**
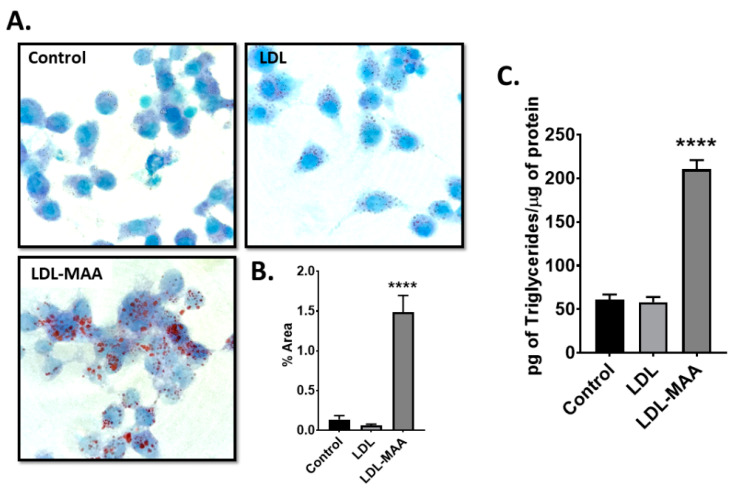
Fat Accumulation in J774 Monocyte/Macrophage Cells. (**A**). Representative 20x microscopic images of J774 cells treated with LDL or LDL-MAA for 24 h and stained with Oil Red O. (**B**). ImageJ quantification of J774 cells treated with LDL or LDL-MAA and stained with Oil Red O. Data is expressed as the mean pixel density of the percent area determined for the entire slide. (**C**). Quantification of triglycerides in control J774 cells and cells treated with LDL or LDL-MAA. Data is expressed as the pictograms of triglycerides per microgram of cellular protein. **** *p* ≤ 0.0001. N = 8.

**Figure 6 ijms-22-12948-f006:**
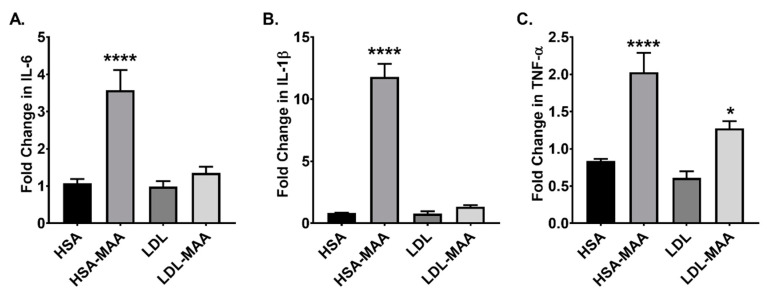
Effects of Treatment of J774 Monocyte/Macrophage Cells with MAA-Modified Proteins. J774 cells treated with HSA, HSA-MAA, LDL, or LDL-MAA and assayed for expression of pro-inflammatory cytokines IL-6, IL-1β and TNF-α. (**A**). IL-6 was significantly increased **** *p* ≤ 0.0001 following HSA-MAA treatment compared to all other groups. (**B**). IL-1β was significantly increased **** *p* ≤ 0.0001 following HSA-MAA treatment compared to all other groups. (**C**). TNF-α was significantly increased **** *p* ≤ 0.0001 following HSA-MAA treatment compared to all other groups. * *p* ≤ 0.05 increased compared to LDL. Data is expressed as the fold change in mRNA compared to treatment with media only. N = 8.

**Figure 7 ijms-22-12948-f007:**
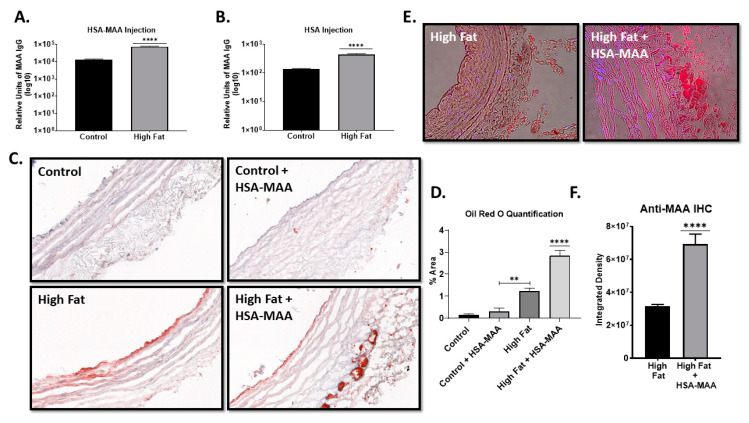
Detection of Antibodies to MAA-Adducted Proteins In Vivo. Male Sprague Daly rats were placed in one of 4 groups: (1) control diet, (2) high cholesterol diet (Western), (3) control diet and injected with HSA-MAA and (4) fed high cholesterol diet and injected IP HSA-MAA. (**A**). Serum antibodies in chow and high cholesterol fed rats immunized with HSA-MAA. Data is expressed as the scientific notation log^10^ in relative units of anti-MAA IgG antibody. (**B**). Serum antibodies in chow and high cholesterol fed rats immunized with HSA. Data is expressed as the scientific notation log^10^ in relative units of anti-MAA IgG antibody. (**C**). Representative 20X Micrographs of sectioned aortas stained with Oil Red O from the 4 groups of rats. (**D**). ImageJ quantification of Oil Red O-stained aortic sections from the 4 groups of rats. Data is expressed as the mean pixel density of the percent area determined for the entire slide. (**E**). Representative 20x IHC images of anti-MAA stained aortic sections from high fat fed compared to high fat fed and HSA-MAA immunized rats. (**F**). ImageJ quantification anti-MAA for IHC images. Data is expressed as the integrated density determined for the entire slide. Significance: ** *p* ≤ 0.001, **** *p* ≤ 0.0001. N = 5 animals per group.

## Data Availability

Data generated in this manuscript is available upon written request to the corresponding author.

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
