# Peer review of "Malondialdehyde-Acetaldehyde Modified (MAA) Proteins Differentially Effect the Inflammatory Response in Macrophage, Endothelial Cells and Animal Models of Cardiovascular Disease"

_ijms, 2021, doi:10.3390/ijms222312948_

Round 1

Reviewer 1 Report

This is an interesting article from Clemens et al., that highlights the impact of high-fat diet on chronic inflammation. Authors have investigated the inflammatory response to malondialdehyde-modified human serum albumin and low-density 24 lipoprotein. They have previously shown that malondialdehyde -adducts are present in aortic atheroma from experimental animals and patients with cardiovascular disease and that antibody titers to these adducts are associated with disease severity and have predictive value. In this paper, they provide a mechanism underlying these observations. However, the manuscript needs to be revised to address the following concerns/comments.

1) All figure legends are unreadable. The axes labels and any text highlighted in the figure should be bigger.

2) It is not required to repeat the description of methods in the results section (e.g. line 103). All along this part needs excessive editing.

3) In Fig 1, is there a way to track the shift from LDL to LDL-MAA? Right now using anti-MAA antibodies, only MAA-linked proteins are observed. If there is a way to track the LDL moiety, reader can actually see the shift.

3) In Fig 2B, the text says SD rats were given normal chow.  In the legend, it says they were given high-fat diet. Which is correct?

4) Isn’t the best comparison in the above experiment be between normal chow vs. high-fat chow for both breeds of rats?

5) In  Fig 6: LDL-MAA showed increased MCP-1 with endothelial cells but not monocytes/macrophages. Is there a possible hypothesis for that?

6) The discussion has a lot of repetition from the Introduction and results. It seems like reading it over again. 

7) You have discussed the difference in signaling between HSA-MAA and LDL-MAA. it will be important to discuss why LDL showed an increase,  but not LDL-MAA?

8) Please carefully read the full text for any omissions.

eg. Rats were fed for 12 weeks; at week, 6 rats were injected with HSA-MAA every week for 5 weeks.

Not clear at what week the rats were injected with HSA-MAA. 

9) The key finding in this paper is a combination of high-fat diet + immunization with HSA-MAA. This is in the abstract, but mixed with all other results. It will be good to highlight this key point.

10) Are there any inhibitors of this pathway that can be used to show that in the presence of this "inhibitor drug", you can prevent accumulation of triglycerides? This will increase the impact of the paper and make this work translatable.

11) A discussion around the impact of this work is essential. 

Reviewer 2 Report

In this manuscript, Clemens et al investigated inflammatory response to MAA-modified proteins (MAA-HSA and MAA LDL specifically) in endothelial cells, macrophages and animal models of CVD and found that these two MAA adducts are immunogenic and can increase fat accumulation in endothelial cell line and macrophages both in mouse and human. They elicit expression of different pro inflammatory proteins in different cell lines, and when combined with high fat diet, can exacerbate aortic fat deposition in rat CVD model in vivo.

There are some questions needed to be addressed before consideration for publication:

1) The authors showed MAA can modify LDL and HSA in vitro, but did not show whether they can detect those adducts in vivo in their rat CVD model. It would also be informative if they can compare MAA adducts between rats with normal diet and western diet;

2) In Figure 5, for completeness of the human data, the authors should probably also quantify fat accumulation and confirm increase in triglycerides for human monocytes, as with mouse monocyte cell line;

3) For in vivo effect, the authors only tested MAA-HSA combined with high fat diet, but not MAA-LDL. Consider that they elicit expression of different proinflammatory proteins and likely different signaling pathways, it would be good to test MAA-LDL as well in this experiment design as well;

4) Though it might be out of scope for this study, it would be nice to perform RNAseq instead of RT-PCR to look for whole transcriptome level change in epithelial cells treated with MAA-proteins and have a more comprehensive view of MAA-adduct elicit response.

Round 2

Reviewer 2 Report

The authors have addressed my concerns.